# Tracking Knowledge Evolution Based on the Terminology Dynamics in 4P-Medicine

**DOI:** 10.3390/ijerph17207444

**Published:** 2020-10-13

**Authors:** Aida Khakimova, Xuejie Yang, Oleg Zolotarev, Maria Berberova, Michael Charnine

**Affiliations:** 1Research Center for Physical and Technical Informatics, Nizhny Novgorod 603098, Russia; aida_khatif@mail.ru; 2School of Management, Hefei University of Technology, Hefei 230009, China; xuejie_y@126.com; 3Russian New University, Moscow 105005, Russia; maria.berberova@gmail.com; 4Institute of Informatics Problems of the FRC CSC, the Russian Academy of Sciences, Moscow 119333, Russia; mc@keywen.com

**Keywords:** cyberspace, virtual reality, 4P-medicine, interlanguage semantic similarity, megalemma, megatoken, informative term, WebVR

## Abstract

The accelerating evolution of scientific terms connected with 4P-medicine terminology and a need to track this process has led to the development of new methods of analysis and visualization of unstructured information. We built a collection of terms especially extracted from the PubMed database. Statistical analysis showed the temporal dynamics of the formation of derivatives and significant collocations of medical terms. We proposed special linguistic constructs such as megatokens for combining cross-lingual terms into a common semantic field. To build a cyberspace of terms, we used modern visualization technologies. The proposed approaches can help solve the problem of structuring multilingual heterogeneous information. The purpose of the article is to identify trends in the development of terminology in 4P-medicine.

## 1. Introduction

The increase in the number of medical publications has made it more important than ever to predict future research trends. Computational modeling of scientific evolution and the tracking of temporary ups and downs of topics are important for financing promising areas of research.

It is becoming increasingly difficult to stay abreast of developments in biomedical science relevant to research. For our research, we used a PubMed (database of medical publications) resource containing many scientific publications [1]. Databases of scientific publications: MEDLINE (National Library of Medicine, USA), Scopus (Elsevier, Netherlands), and Web of Science (Clarivate Analytics, USA) differ in the subject matter and the toolkit they provide. MEDLINE focuses primarily on biomedical disciplines, while Scopus and Web of Science are multidisciplinary. The MEDLINE was created and maintained by the US National Library of Medicine (Rockville Pike, Bethesda MD, USA). The database is updated weekly and almost completely covers all medical journals in the world. The work with the information array of the database is carried out using the PubMed search engine, which operates on the same server as the database itself. PubMed is the largest database of scientific publications on medicine. Figure 1 shows the growth rate of scientific publications in the field of 4P-medicine from 2000 to 2019 using available data from PubMed. 4P-medicine (Predictive, Personalized, Preventive, Participatory) is an ideology which focuses on an individual approach to a patient. Its purpose is a preclinical detection of diseases and the development of a set of preventive measures. 

With the creating of new terminology, we need to unify new terms and prescribe certain meanings to lexical units. For example, the term “gene” had several different meanings during the last century. Since 1960, the term “gene” has meant an abstract “unit of inheritance”. Then, it meant a linear segment in the chromosome, and some time later, scientists described it as a linear segment in a DNA molecule. 

Further experimental studies led to the refinement of the value according to Portin and Wilkins [2]. It turned out that the components of a gene are not always contiguous.

The evolution of the term’s meaning may be attributed to the success of scientific research. For several decades, Hidradenitis suppurativa was known by many terms as histopathologic discoveries were made [3]. 

Scientists have been examining the dynamics of the development of AIDS-related terminology for many years. In this case, the development of science caused a change in terminology, but it was also socially determined. We have seen a change in terminology in this area. Until 2008, the term “victims of HIV” was used more often in medicine. Scientists used the terms “positively infected” and “negatively infected” in the early 1990s, and then, these terms fell out of use [4]. 

The constant change in medical terminology reflects real advances in this area [5]. 

There is a clear trend towards expanding terminology replenishment tools, including derivation and the formation of combinations of terms. Many scholars believe that the internationalization of terms is an effective tool for language development. The use of international terms allows filling of the lacunae in national terminology with more abstract vocabulary lexical units. The organization and presentation of knowledge (a knowledge base) is a central problem of new information technologies [6]. Guo et al. [7] presented a model for describing and predicting key features of new research areas. They showed that a sudden increase in the frequency of specific words is one of the signs indicating the formation of a new field of research.

When searching for texts, it is important to determine which topic the document relates to. This problem can be solved by thematic modeling (TM), which allows the building of models of a collection of text documents. Thematic modeling has a significant history of application in studies of the dynamics of the development of scientific trends. The existing models are mainly based on the latent Dirichlet distribution (LDA) thematic model [8]. LDA is a generative process that models each document as a mixture of topics, where each topic corresponds to a polynomial distribution of words. LDA was used to detect various research topics from a corpus of scientific papers [9,10]. In these studies, scientific ideas and areas were modeled as word distributions. He et al. [11] proposed a topic model by adapting the latent Dirichlet distribution of the model to a citation network to develop approaches to assessing the evolution of topics based on citation. The authors presented an iterative structure for teaching a topic based on a citation network. Experimental results have shown that the approach allows tracking of the evolution of a topic in a large dataset.

Along with the rapid development of topic modeling in machine learning, many LDA extensions have emerged.

Rosen-Zvi et al. presented the author–topic model (ATM), where a document is modeled as a product of a mixture of authors topics without temporal ordering [12]. Bolelli et al. proposed a segmented author–topic model (S-ATM) based on the ATM model. It integrates the temporal characteristics of a collection of documents into a generative process [13]. The S-ATM shows the ability to identify the evolution of topics over time.

The dynamic topic model (DTM) is designed to track the evolution of a topic by sequentially grouping a set of documents based on the assumption that topics in the current time interval have evolved smoothly from the corresponding topics in the previous time interval [14,15,16].

Thematic modeling has been used in medical and biological sciences. Chen et al. [17] proposed a biological dynamic topic model (Bio-DTM). Topics such as biosynthesis of ginsenoside, cultivation of ginseng, etc., have been derived from scientific articles on the subject of “Ginseng” using a Bio-DTM. The most frequently occurring words have been highlighted for each topic. ThemeRiver was used to visualize the evolution of themes in 16 time intervals [18].

There were proposed methods to explore new trends using word frequency analysis while tracking the frequency of keywords/phrases over time.

Asooja et al. [19] proposed regression models to predict the growth of a scientific topic as a temporary distribution of keywords in the future. They generated the dataset from all Language Resources and Evaluation Conferences [20]. The dataset consisted of a temporal estimate of the tf-idf evolution of various keywords in conferences. Modeling the temporary development of topics made it possible to identify new trends in conferences in the field of computational linguistics. Wu et al. [21] investigated the development priorities and research directions in the field of mental disorders, analyzing the frequency of keywords using the Sci2 visualization tool [22]. 

Keywords also provide insights into historical trends. The joint occurrence of terms makes it possible to determine the most frequently encountered phrases in the texts of articles [23,24]. One of the first attempts to generalize a large set of documents for visualization, to understand topics or trends, was suggested by Voegele [25,25]. Because of the growing amount of information, modern medicine cannot do without the latest technologies such as machine learning and data mining [26]. Analysis of innovative approaches in medicine shows that today, the processing of large amounts of data is impossible without the formation of knowledge bases through the study of historical medical data. Case-based reasoning (CBR) systems are very useful in medicine. The use of similar systems for the early detection of breast cancer were based on disease feature ranking. CBR systems provide physicians with valuable information, including historical disease data. Based on historical data, Gu [27] proposed a weighted heterogeneous value distance metric with a genetic algorithm, which is very meaningful for enriching the methodologies of case-based knowledge discovery. The use of artificial intelligence systems in modern medicine makes it possible to diagnose diseases more accurately at an early stage. The use of modern technologies such as cloud computing and artificial intelligence permitted Gu to create a data-driven intelligent platform called CBHKS [28,29].

Unlike most published approaches, the proposed approach defines each keyword with different meanings by different researchers (according to their personal understanding) with the compatibility of words when we take into account combinations of terms with their surroundings. We include word combinations in clusters. To find the terms, we did not use the author keywords. We applied alternative sources of terms, such as headings and annotations. For their identification and statistical evaluation, we applied an automatic approach. Medical information systems store a large amount of poorly structured data. Health data have various formats and are extracted from many sources using different terms. Because of heterogeneous formatting and scattered terminology, big medical data provide too few options for data analysis and decision support systems.

The main problems in creating a centralized knowledge base are the semantic and syntactic heterogeneity of health data. Multilingual medical terminology complicates the process of integrative cognition. The closer the terms by context, the closer they are in a semantic space. One of the new effective visualization tools is WebVR [30].

We propose a three-dimensional space of scientific terms like the Chen constellation. However, our cyberspace has improved visualization features, including ranking of relevant terms and semantic clustering [31]. Because of the growing amount of new information, it is becoming increasingly difficult to process and generalize it. Modern research in the field of personalized medicine (PM) examines individual research areas, forming dictionaries of medical terms. Ali-Khan et al. created a collection of terms related to personalized medicine [32]. We offer a universal ontological approach based on the use of bibliometric methods of analysis and methods of intellectual processing of unstructured information. Research in this area is aimed at analyzing large amounts of information (big data). There is no clear and widely agreed-upon definition of PM, although the international community has shown a growing interest in this topic. We propose a new approach to the construction and development of terminology in 4P-medicine, based on the study of the dynamics of changes in medical terminology. We offer a more general mechanism for analyzing medical data. We propose a three-dimensional space of scientific terms like the Chen constellation. We present a new method for assessing semantic similarity. We propose to evaluate the information content of text objects as a concentration of ideas. An idea is a combination of meaningful terms. Future research will be more successful if modern methods of processing, structuring, and visualizing large amounts of information are applied.

The rest of the paper is organized as follows: The next section introduces the data for research and the methodology we used in this study. Section 3 introduces megalemmas and a method of their construction. Section 4 presents findings from the study.

Regarding the novelty of the project, an automatic analysis of trends and detection of new terms in large volumes of scientific publications (Big Data) in the field of 4P-medicine were performed using free scientific libraries. The results of the project are not tied to a specific subject area and can be used in various fields of activity.

## 2. Materials and Methods

We present an algorithm for identifying trends in the development of terminology in the field of 4P-medicine. First, a dictionary of key terms of the subject area is created. Then, a temporary dictionary of new terms is created. At the next step, an array is created to collect statistics on the term’s frequency (Formula (1)).
S = {D^e^_i_{D^n^_k_, R_k_}}(1)

Here, D^e^_i_ is an element of the term’s vocabulary, D^e^_i_ is an element of the temporary dictionary of new words. R_k_ is a frequency of occurrence of a new word D^n^_k_ next to the key term D^e^_i_. If a new word is found in the vicinity of the keyword, then it is included in the temporary dictionary, the frequency of its occurrence increases, and a new phrase is built. Next, phrases with a frequency of occurrence above the threshold are selected. After building a temporary dictionary, the expert makes a decision to include a new term in the main dictionary.

The terminology development trends are calculated according to the following formula:(2)$$Trend_{W_{i}Y_{i+1}Y_{i}}=\frac{\left(N_{W_{i}Y_{i+1}}+0.1\right)/\left(NSUM_{Y_{i+1}}+0.1\right)}{\left(N_{W_{i}Y_{i}}+0.1\right)/\left(NSUM_{Y_{i}}+0.1\right)}$$
where $N_{W_{i}Y_{i}}$ is the number of articles with the word $\;W_{i}$ per year $Y_{i}$, and a $NSUM_{Y_{i}}$ is the total number of articles published per year $Y_{i}$.

We built the cyberspace of scientific terms in the field of 4P-medicine using interactive 3D graphics in WebVR. Cyberspace can be a useful tool for integrating heterogeneous information. The cyberspace approach provides the ability to visualize multilingual terms in one semantic field [33].

The heterogeneity of medical data from various sources complicates the task of their integration. The proposed semantic cyberspace can help with integrating data and knowledge for biomedical research. We consider an assumption that aspects of medicine include similar ideas, represented by sets of terms.

We used the word2vec method to identify a semantic environment of terms and a semantic similarity of documents, and we applied WebVR methods for three-dimensional visualization of the calculation results.

We extracted articles related to 4P-medicine from the PubMed database with terms “predict” and “personalis(z)e” in the headings and abstracts. These terms had the most numerous derivatives and collocations. In addition, we chose the terms “prognosis” and “prevent” for the experiment.

We treat the megalemma as one word. The nominal group consists of a word consistent with the determinants of gender, number, and case. The genetic group includes two nominal groups. Megatoken is the sequence of megalemmas for each genetic group listed in alphabetical order. Thus, the collocations correspond to one megatoken [GEN + DISEASE].

To build a cyberspace, we must categorize terms. We identified three subgroups for numerous groups of terms related to “predict” and “personalis(z)e”. The first group included derivatives. The second group incorporated megatokens. The third group included independent terms that did not form megatokens. We consider the first and the second subgroups as categories.

A total of 172 elements formed the most numerous groups with the root “predict”, which consisted of (1) 12 derivatives (Table 1), (2) 30 megatokens (Table 2), and (3) 56 independent collocations.

Table 1 shows that derivatives with the root “predict” appeared widely before 2007. Thus, the relative growth in the use of the term “predict” in 2019 compared with 2007 was only 0.79%. In the last decade, the derivatives “unpredictable” and “unpredictability” have come into use.

Table 2 shows that many megatokens (relative increase of 100%) appeared only in the last decade: (PREDICT + PREVENT, PREDICT + CLINIC, PREDICT + PERFORM, PREDICT + DISEASE, PREDICT + POSITIVE, PREDICT + ERROR, PREDICT + TOOL, PREDICT + TOOL, PREDICT + TOOL UNIQUE, PREDICT + PATIENT, PREDICT + DEVELOP, PREDICT + DIAGNOSE).

Other collocations that have not yet formed megatokens: medicine predictive, predict changes, predict efficacy, predict future, predict onset, predict overall, predicted increases, predicted lower, predicted neuroticism, predicting long-term, prediction postoperative, prediction score, prediction using, predictive capability, predictive control, predictive index, predictive power, predictive relationship, predictive utility, predictive validity, predictors burnout, predictors depression, predictors moderators, predictors successful, predictors suicide, prevalence predictors, reliable predictive, and others.

The developed application based on the word2vec method identifies random phrases, depending on the size of the “window” among the text. The last subgroup consists of such random combinations of words as “also predicted”, “predicted high”, “can be predicted”, “forecast used”, and so on. Obviously, we do not consider such combinations as “forming cyberspace”.

To build a cyberspace, we chose (1) derivatives with “predict” as the root word (purple spheres); and (2) significant collocations with derivatives of “predict” as the root word (green spheres).

The second group consists of 85 lexical units with “personalized” as the root word. The group included 8 derivatives (Table 3), 10 megatokens (Table 4), and 41 independent collocations. Random phrases such as “towards personalized”, “using personalized”, “based on personalized”, and so on are the result of the size of the word2vec method “window”.

Table 3 shows the derivatives “personalization” and “personalisation” formed after 2007. 

Table 4 shows some megatokens (whose growth was 100%) formed only in the last decade: (PERSONALIS(Z)E + PREDICT, PERSONALIS(Z)E + CARE, PERSONALIS(Z)E + MODEL, PERSONALIS(Z)E + APPLICATION).

There are other collocations (not included in megatokens): contribute personalized, guide personalized, towards personalized, effective personalized, era personalized, using personalized, improve personalized, used personalize, based personalized, facilitate personalized, contribute personalized, toward personalized, enable personalized, provide personalized, future personalized, designing personalized, personalized risk, personalized management, personalized cancer, personalized interventions, personalized precision, design personalized, personalized drug, tool personalized, personalized pain, response personalized, and more.

To build a cyberspace, we chose (1) derivatives with “personalis(z)e” as the root word (black spheres); and (2) significant collocations with derivatives having “personalis(z)e” as the root word (blue spheres).

Thus, we obtained four types of spheres for location in cyberspace.

We examined the dynamics of terms that are derivatives and phrases from the root word “prognosis” (Table 5) and the root word “prevention” (Table 6) to complement the general picture in the field of 4P-medicine (prognostic, preventive, personalized, and participatory). Because of the small number of derivatives and collocations, we did not divide them into subgroups.

To build a cyberspace, we chose (1) terms and collocations with “prognosis” as the root word (red spheres); and (2) terms and collocations with “prevent” as the root word (yellow spheres).

Thus, we got six categories of spheres for location in cyberspace: (1) derivatives with “predict” as the root word; (2) megatokens with “predict” as the root word; (3) derivatives with “personalis(z)e” as the root word; (4) megatokens with “personalis(z)e” as the root word; (5) terms and collocations with “prognosis” as the root word; and (6) terms and collocations with “prevent” as the root word. 

## 3. Results

Figure 2 shows the research results. Words and collocations obtained from “prognosis” and “prevent” came into use after 2007. Therefore, their total number in the collection is less than the number of words and collocations derived from “predict” and “personalis(z)e” (which were widely used until 2007).

Figure 2 shows the rise in popularity of terms related to 4P-medicine in scientific publications since 2007. The number of publications related to the term “predict” is the maximum (blue column). The maximum dynamics in increases popularity were shown by the term “prognosis”. The terminology related to predictive medicine has the greatest upward trend.

We used WebVR technology to create a three-dimensional visual map of the terminological set that characterizes the field of 4P-medicine. The spheres in Figure 3 are scientific terms. The application calculated the sizes and coordinates of the spheres automatically using the methods mentioned above. WebVR technology provides the ability to create a three-dimensional model, rotate it, and present a view of it from different angles [33].

Figure 3 reflects a three-dimensional constellation of terms for 4P-medicine. 

Figure 3 demonstrates a new way of displaying connected vertex networks based on A-Frame technology. The application calculated the sizes and coordinates of the spheres automatically using the methods mentioned above. WebVR technology provides the ability to create a three-dimensional model, rotate it, and present a view of it from different angles [34].

This article discusses the visualization of a collection of terms related to 4P-medicine. The collection includes 8067 English terms and collocations. To discover the terms and phrases, we used sets of articles from the PubMed database for the period from 2007 to 2019. We chose this period because the European Association for Predictive, Preventive, and Personalised Medicine appeared in 2008, and it actively promotes the ideology of 4P-medicine.

Figure 3 shows the resulting three-dimensional visual map of terms. The higher a term’s rating, the larger the size of the sphere [35]. Semantically related terms form clusters, as shown in Figure 3.

## 4. Discussion

We developed an application for two languages: English and Russian. Despite the difference in their structures, it was possible to establish an interlinguistic correspondence between them based on the analysis of the semantic environment of the terms [36]. 

The basis of our approach was the transformation of text into a set of megalemmas. Megalemmas are language-independent. For the research data, we used machine translation methods.

“Disease” can be seen as a megalemma with the words “bolezn”, “bolet”, “bolnoy”, “bolnitza”, and “bolnichnye” and includes the English words “disease”, “diseased”, and “diseases”. The “virus” megalemma contains the Russian words “virusnye” and “virusologia” and the English words “virus” and “viruses”. The megalemma “gene” contains the Russian words “gennye”, “genetica”, “genom”, and “genomica” and the English words “gene”, “genetic”, “genomic”, and “genome”. A megalemma usually corresponds to one word in the text. However, nominal and genetic groups are more informative. The sequence of megalemmas for each genetic group listed in alphabetical order is called a megatoken. Thus, collocations such as “genetic disease” correspond to one megatoken [GEN + DISEASE].

Replenishment with new words and terms and their translations will improve the dictionary of megalemmas and megatokens. The updated dictionary will provide increased accuracy in measuring multilingual semantic similarities. New multilingual words can form a new megalemma if they have a similar context of megalemmas in parallel texts, and their semantic vectors (Word2vec) are also very similar [37].

Analysis of the PubMed collection of articles revealed the following collocations in the field of medical genetics: genomic and proteomic methods, gene knockout technology, gene therapy, gene technology, genetic profile, genomics, gene correction methods, molecular genetics, genetic engineering methods, genetic testing, gene diagnostics, genetic screening, postgenomic technology, and gene delivery. This list differs from the terminology of the “gene” category presented in MeSH [38].

Then, we selected the closest pairs of terms for the key term “patients” in articles from 2007 to 2019. We list 145 collocations of terms in descending order of rating (see Appendix A). 

From the word pairs obtained by the word2vec method, we chose meaningful collocations in the next stage. We saw that most collocations refer to patients and their diseases (for example, chondrosarcoma patients, cirrhotic patients, asthma patients, melanoma patients). Some collocations refer to patient characteristics (e.g., high-risk patients, Chinese patients, female patients). These last collocations we can be included in the megalemma’s dictionary.

We selected combinations of terms that indicated patients and their diseases (21.9%). These phrases represent megatoken DISEASE + PATIENT (see Appendix B). 

Using the number of terms, we identified the roots “predict” and “personali” by year in aggregate, and plotted a graph to determine the trend of publication activity from 1975 to 2018 (shown in Figure 4).

The polynomial trend (degree = 3) describes the data with a very high degree of approximation, equal to 0.9133. According to calculations in accordance with the trend, for 2019, the value of the number of publications is 1940. Upon entering in PubMed “predict [Title]) OR personali [Title]”, we received 1644 publications. The standard deviation was calculated to be ±154. Therefore, we are almost within the acceptable range of values, and our methods of highlighting key terms allow us to predict publication activity with a high degree of probability.

We analyzed the use of the word treatment with all derivatives (predictor(s), predictive, predict(ing, ed), prediction). Table 7 contains the currently used phrases. Probably, we should expect the appearance of phrases: predictive risk/error; predictive tool (s); accurate predictor(s); personalized/individualized predictor(s); unique predictor(s); predict relationship; significant/important prediction; successful/reliable prediction; unique prediction.

Figure 5, Figure 6, Figure 7 and Figure 8 demonstrate the dynamics of collocation formation during past years.

The derivative “predictor(s)” began to be used with the words “important”, “clinical”, “successful”, “suicide”, “negative(ly)”, and “depression”. The word “treatment” has disappeared from use. The use of the words “significant”, “identify(ing, ied)”, and “response” has reduced (shown in Figure 5).

The derivative “predictive” began to be used with the words “model(s)”, “prevalence”, “accuracy(ate)”, “performance”, “personalized”, “develop(ment)”, “role”, and “diagnostic(s)”.

The use of the words “value(s)”, “biomarker(s)”, “power”, “validity”, and “factor(s)” has reduced. The use of the word “prognosis(tic)” has grown (shown in Figure 6).

The derivative “predicting(ed)” came to be used with the words “risk”, “accura(cy, ate)”, “clinical”, “biomarker(s)”, “patient”, “positive”, and “disease”.

The use of the words “significant” and “individual(ized)” has reduced. The use of the words “response”, “model(s)”, “outcome”, “factor(s)”, and “variables” has grown (shown in Figure 7).

The derivative “prediction” came to be used with the words “personalized”, “survival”, “clinical”, “tool(s)”, “error”, and “disease”.

The use of the word “risk” has reduced. The use of the words “response”, “model(s)”, and “accur(acy, ate)” has grown (shown in Figure 8).

Based on their proximity, we defined the terms with derivatives of the roots “predict” and “personalis(z)e” (shown in Figure 9 and Figure 10).

The ranking of terms that have appeared in the vicinity of the derivatives “predict” and “personalis(z)e” in the last decade shows that 4P-medicine scientists pay special attention to immunological aspects (immune, immunotherapy), using machine learning methods, storage methods, and information processing (PsycINFO (database of abstracts of literature in the field of psychology, American Psychological Association), algorithm, area under the curve (AUC)). The scientific community has recognized the need for a 4P-approach in the treatment of diseases such as CRC (colorectal cancer), HCC (hepatocellular carcinoma), OS (Osgood–Schlatter disease), PFS (post-finasteride syndrome), bladder cancer, and MS (multiple sclerosis). The study of nucleic acids (miRNAs, CtDNA (circulating tumor DNA)) is especially important (shown in Figure 9). 

As a result of the study, the most promising areas of development in the field of 4P-medicine were identified. We can use such phrases and terms to evaluate the trends in the worldwide extension of chronic diseases.

## 5. Conclusions

We proposed the cyberspace of the significant terms related to 4P-medicine, implemented by interactive three-dimensional graphics in WebVR.

Selected articles from the PubMed database related to 4P-medicine with the terms “predict”, “prevent”, “prognosis”, and “personalis(z)e”. The terms “predict” and “personalis(z)e” had the most numerous derivatives and collocations.

To build a cyberspace, we divided the terms into categories. For the most numerous terms “predict” and “personalis(z)e”, we identified four categories to build a cyberspace, including derivatives and megatokens. We excluded random collocations and collocations that do not form megatokens. To complement the general picture in the field of 4P-medicine, we added to cyberspace derivatives and collocations from the roots of “prognosis” and “prevention”. The cyberspace represents a collection of scientific terms.

In addition, we identified megatokens for the last decade, such as PREDICT + PREVENT, PREDICT + CLINIC, PREDICT + PERFORM, PREDICT + DISEASE, PREDICT + POSITIVE, PREDICT + ERROR, PREDICT + TOOL, PREDICT + UNIQUE, PREDICT + PATIENT, PREDICT + DEVELOP, PREDICT + DIAGNOSE, PERSONALIS(Z)E + PREDICT, PERSONALIS(Z)E + CARE, PERSONALIS(Z)E + MODEL, and PERSONALIS(Z)E APPLICATION.

The dictionary of megatokens was created with collocations obtained by analyzing collections of articles. From the PubMed database, we extracted the following collocations related to medical genetics, such as gene technology, genetic testing, genetic screening, gene delivery, genetic profile, genetic engineering methods, genomic and proteomic methods, change in gene expression, gene therapy, postgenomic technology, technology knockout genes, and gene correction methods. These combinations differ from the terminology presented in MeSH.

We found that only 21.9% of all collocations received by the key term “patients” refer to patients and their diseases. We can represent these collocations in the form of megatoken DISEASES + PATIENT: patients with CRC (colorectal cancer), patients with PD (Parkinson’s disease), patients with CAD (coronary artery disease), patients with tumors, patients with NSCLC (non-small-cell lung cancer), patients with TNBC (triple negative breast cancer), and so on. Other collocations referred to the characteristics of the patient (for example, high-risk patients, patients from China). We included them in the megalemma’s dictionary.

Such semantic constructs as megalemmas and megatokens provide the ability to convert multilingual texts into similar constructions, independent of the language. We can improve the correctness of evaluating the interlanguage semantic similarity using these constructs. Besides, the temporal dynamics of these constructs demonstrates the evolution of the scientific area.

Therefore, the terms used most often in 4P-medicine (from those that appeared earlier) are target(ing, ed), target(s); model(s), modeling; identify(ied, ing); prognosis, prognostic; accuracy, accurate; biomarkers; gene(s), genetic; cancer(s); lung cancer; and radiotherapy (shown in Figure 10).

Thus, we can draw the following conclusions:

The field of 4P-medicine focuses on such diseases as CRC (colorectal cancer), HCC (hepatocellular carcinoma), OS (Osgood–Schlatter disease), PFS (post-finasteride syndrome), bladder cancer, MS (multiple sclerosis), cancer(s), lung cancer, diabetes, bipolar disorder, prostate cancer, borderline personality disorder (BPD), breast cancer, and Parkinson’s disease (PD).

The area of 4P-medicine actively uses methods of statistics, storage, and processing of information (machine learning, PsycINFO database, algorithm, area under the curve (AUC), confidence interval, logistic regression, and regression analysis).

Important functions and parameters of the predictive aspect of 4P-medicine are targeting, modeling, accuracy, prognosis, imaging, testing, significance, precision, and risk.

Widely used methods are radiotherapy, chemotherapy, and immunotherapy. 

Actively used methods for diagnostics are screening, magnetic resonance imaging (MRI), biopsy, and biomarkers. 

Methods of medical genetics are miRNAs, CtDNA (circulating tumor DNA), gene(s), genetic, genome, and epigenetic.

Today, mental health is an important area of research. The following terms were detected: (behavior(s), antidepressant, suicid(e, al), mental health, psychological, distress, neuroticism, stress, depressive).

As a result of the analysis, trends in the development of new directions (terminology) in the field of 4P-medicine were identified. A neighborhood approach was used to identify these trends. There were terms that were defined that are gradually falling out of use and terms whose popularity is growing. 

All these areas are developing very actively.

This article, based on statistical analysis, allows us to draw a conclusion about the most demanded areas of diseases and the dynamics of the development of terminology in the field of 4P-medicine.

As a result of this work, a mechanism was developed for identifying trends in the subject area (4P-medicine). This can already provide significant assistance in the formation of strategic plans for the development of various areas of medicine.

It is planned to further expand the proposed methods of searching for the most popular and, accordingly, developing research methods associated with trends in the development of diseases in the field of 4P-medicine. For this, not only statistical, but also semantic mechanisms will be developed to highlight the trends of the most popular research methods in the framework of new directions in 4P-medicine. To do this, we have mechanisms that allow highlighting of the connections between terms and the associated processes. In the future, we intend to develop mechanisms for the automatic construction of ontologies based on the analysis of full-text scientific publications.

## Figures and Tables

**Figure 1 ijerph-17-07444-f001:**
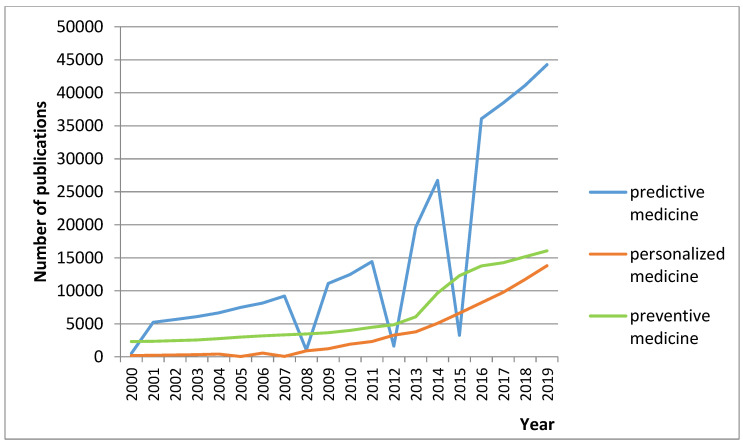
Absolute number of publications in the PubMed database from 2000 to 2019 received by a title/abstract search using the terms “predictive medicine”, “personalized medicine”, and “preventive medicine” [1].

**Figure 2 ijerph-17-07444-f002:**
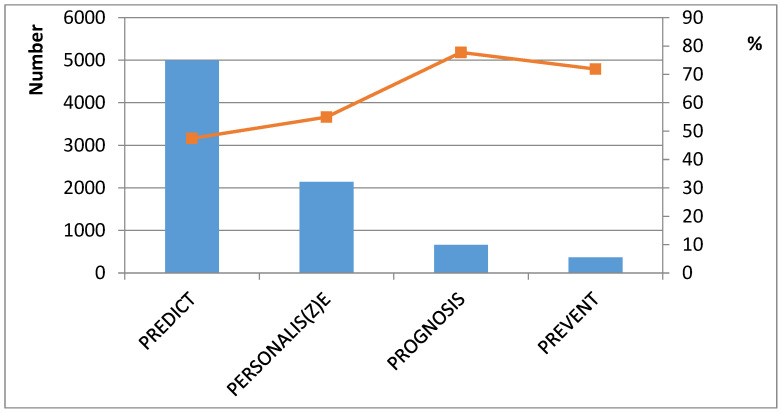
The number and relative growth of derivatives and collocations with the root words “prognosis”, “prevent”, “predict”, and “personalis(z)e” since 2007.

**Figure 3 ijerph-17-07444-f003:**
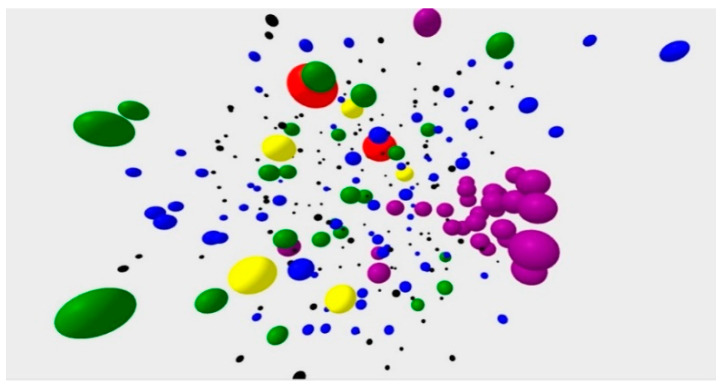
A-Frame technology for 3D visualization.

**Figure 4 ijerph-17-07444-f004:**
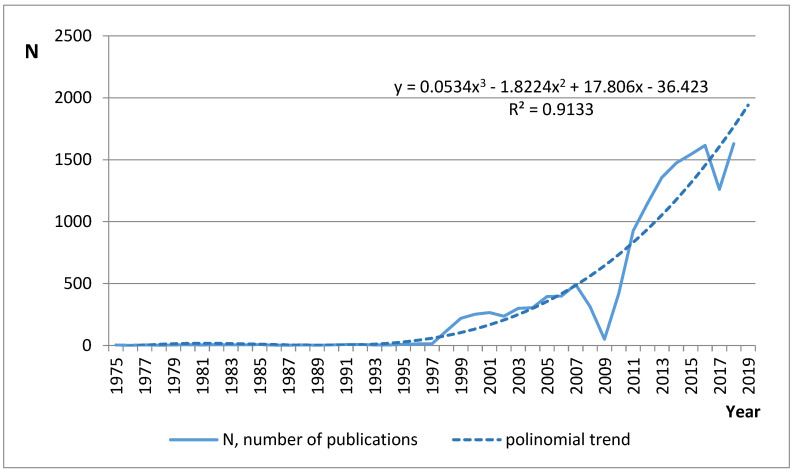
The trend of publication activity from 1975 to 2018 and forecast for the future.

**Figure 5 ijerph-17-07444-f005:**
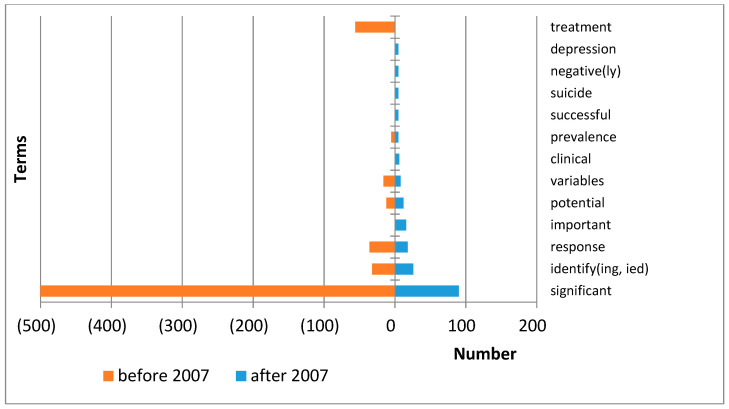
The frequency of occurrence of terms in the vicinity of the derivative “predictor(s)” over the past decade.

**Figure 6 ijerph-17-07444-f006:**
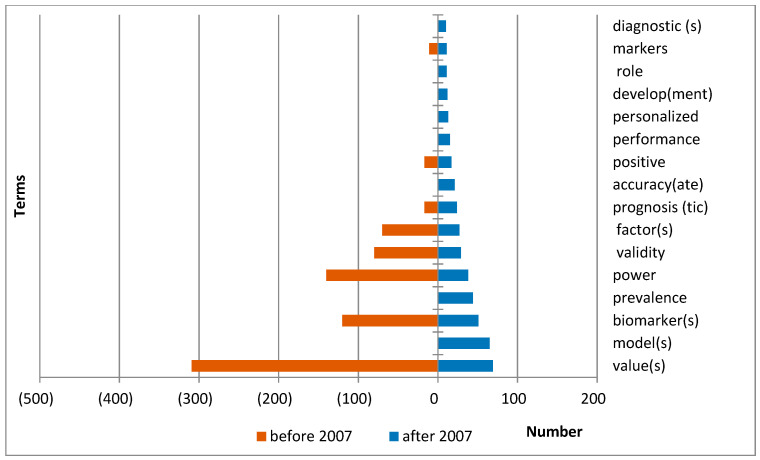
The frequency of occurrence of terms in the vicinity of the derivative “predictive” over the past decade.

**Figure 7 ijerph-17-07444-f007:**
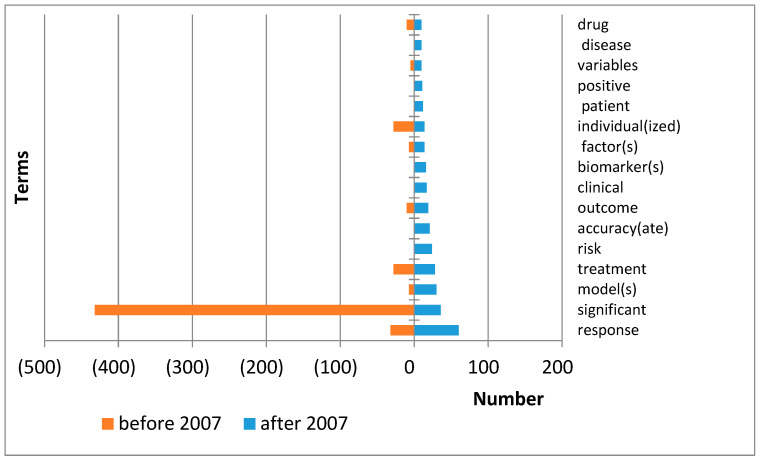
The frequency of occurrence of terms in the vicinity of the derivative “predicting(ed)” over the past decade.

**Figure 8 ijerph-17-07444-f008:**
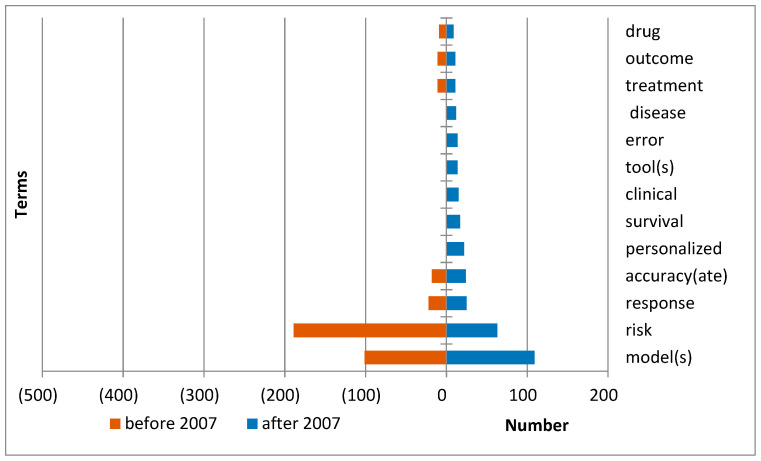
The frequency of occurrence of terms in the vicinity of the derivative “prediction” over the past decade.

**Figure 9 ijerph-17-07444-f009:**
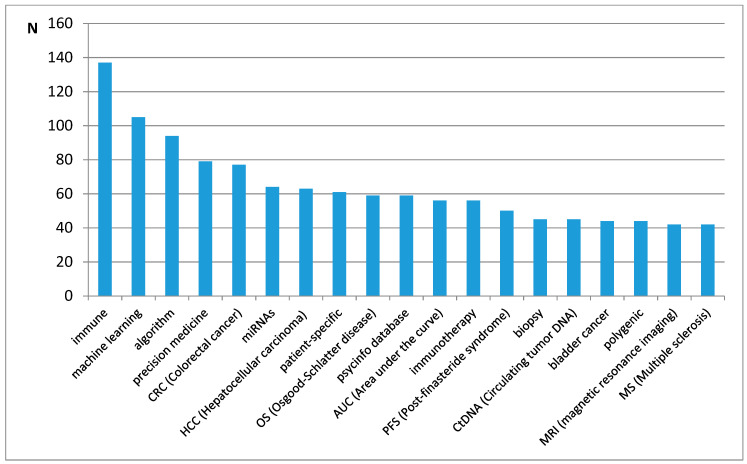
The frequency of occurrence of terms in the vicinity of the derivatives “predict” and “personalis(z)e” over the last years (N is the number of terms).

**Figure 10 ijerph-17-07444-f010:**
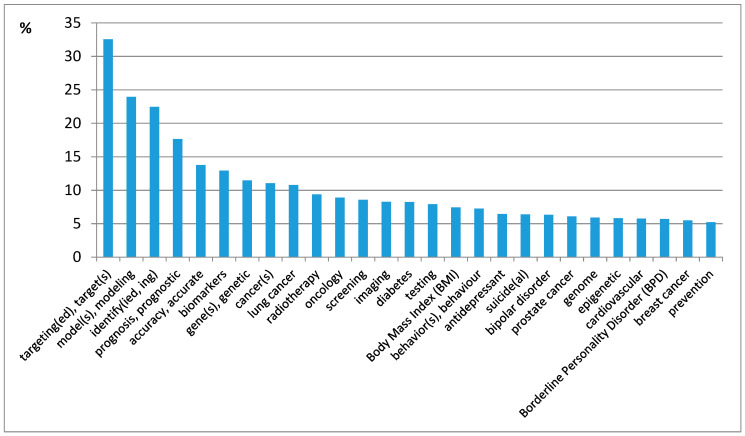
Ranking of terms that appeared before 2007 by relative growth over the last years (N is the relative growth).

**Table 1 ijerph-17-07444-t001:** Statistical results for derivatives with “predict” as the root word, extracted from the PubMed database for the period from 2007 to 2019 (purple color in cyberspace).

No.	Derivatives	Number of Appearances	Relative Growth from 2007, %
1	predictive	629	0.79
2	predict	609	0.66
3	prediction	594	0.95
4	predicted	586	0.44
5	predictors	493	0.57
6	predicting	313	1.35
7	predictor	208	1.33
8	predicts	125	3.03
9	predictions	110	5.26
10	unpredictable	19	100.00
11	predictability	19	33.33
12	unpredictability	5	100.00

**Table 2 ijerph-17-07444-t002:** Key collocations with derivatives that have the root “predict” extracted from the PubMed database for the period from 2007 to 2019 (green color in cyberspace).

No.	Megatoken	Collocations	Amount	Relative Growth from 2007, %
1	PREDICT + MODEL	prediction model(s), predictive model(s), predictive modeling, models predicting, model predict(ive/ed/ing/ions)	204	73.53
2	PREDICT + SIGNIFICANT	significant predictive, significant predictor(s), significantly predicted	132	11.28
3	PREDICT + RESPONSE	predict response(s), predicted response, predict(ing/ion/ors) response, response prediction	97	57.79
4	PREDICT + RISK	predict(ing) risk, risk prediction	87	45.69
5	PREDICT + VALUE	predictive value(s)	69	14.43
6	PREDICT + ACCURATE	accurate prediction, accurately predict(ed), prediction accuracy, predictive accuracy	66	86.36
7	PREDICT + TREAT	predict(ing/ion/ive/or/ors) treatment	63	35.39
8	PREDICT + PREVENT	preventive predictive, predictive preventive	44	100.00
9	PREDICT + FACTOR	factors predicted, factors predicting, predictive factor(s)	41	51.74
10	PREDICT + CLINIC	clinical prediction, predict(ing/ion/or) clinical	38	100.00
11	PREDICT + IDENTIFY	identify predictive, identify(/ied/ing) predictors	31	65.59
12	PREDICT + OUTCOME	outcome prediction, predict outcome(s)	30	65.00
13	PREDICT + PERFORM	predictive performance, prediction performance	30	100.00
14	PREDICT + NEGATIVE	negative predictive, negative predictor, negatively predicted	27	74.07
15	PREDICT + TRAIT	traits predict (ed), traits predicting	24	72.22
16	PREDICT + DISEASE	predict(ion) disease, disease prediction	22	100.00
17	PREDICT + POTENTIAL	potential predictive, potential predictors	20	70.00
18	PREDICT + INDIVID	individualized prediction, predict individual	20	53.34
19	PREDICT + DRUG	predict(ion) drug	19	50.00
20	PREDICT + PROGNOSIS	prognosis prediction, prognostic prediction, prognostic predictive, predict prognosis, predictive prognostic	18	83.33
21	PREDICT + POSITIVE	positively predicted, positive predictive	18	100.00
22	PREDICT + ROLE	predictive role, role predicting	18	80.56
23	PREDICT + VARIABLE	predictor variables, variables predict(ed)	18	56.48
24	PREDICT + IMPORTANT	important predictor(s)	16	81.25
25	PREDICT + ERROR	prediction error(s)	14	100.00
26	PREDICT + TOOL	prediction tool(s)	14	100.00
27	PREDICT + UNIQUE	unique predictive, uniquely predicted	14	100.00
28	PREDICT + PATIENT	predict(ing) patient	12	100.00
29	PREDICT + DEVELOP	develop predictive, development predictive	12	100.00
30	PREDICT + DIAGNOSE	diagnostic predictive, predictive diagnostics	10	100.00

**Table 3 ijerph-17-07444-t003:** Statistical results for derivatives with “personalis(z)e” as the root word extracted from the PubMed database for the period from 2007 to 2019 (black color in cyberspace).

No.	Derivatives	Number of Appearances	Relative Growth from 2007, %
1	personalized	1039	0.84
2	personalised	131	16.67
3	personalize	58	16.67
4	depersonalization	56	3.7
5	personalization	47	33.33
6	personalizing	42	33.33
7	personalisation	15	100
8	personalise	10	100

**Table 4 ijerph-17-07444-t004:** Significant collocations with derivatives having “personalis(z)e” as the root word extracted from the PubMed database for the period from 2007 to 2019 (blue color in cyberspace).

No.	Megatoken	Collocations	Amount	Relative Growth from 2007, %
1	PERSONALIS(Z)E + MEDICINE	medicine personalized, personalis(z)ed medicine, personalized medical	237	16.78
2	PERSONALIS(Z)E + TREATMENT	personalis(z)e(d) treatment(s), personalizing treatment, treatment personalization	168	55.16
3	PERSONALIS(Z)E + THERAPY	personalized therapy, personalized therapeutic, personalized therapies	63	35.45
4	PERSONALIS(Z)E + DEVELOP	develop(ing/ed) personalized, development personalized	56	49.29
5	PERSONALIS(Z)E + PREVENT	preventive personalis(z)ed, personalized preventive, personalized prevention	47	100
6	PERSONALIS(Z)E + APPROACH	personalized approach(es), approach personalized	42	61.90
7	PERSONALIS(Z)E + PREDICT	personalized predict(ion, ive), predictive personalized, prediction personalized	35	100
8	PERSONALIS(Z)E + CARE	personalis(z)ed care, personalized healthcare	26	100
9	PERSONALIS(Z)E + MODEL	personalized model(s), models personalized	20	100
10	PERSONALIS(Z)E + APPLICATION	application(s) personalized, application personalized	11	100

**Table 5 ijerph-17-07444-t005:** Terms and collocations with “prognosis” as the root word (red color in cyberspace).

No.	Terms and Collocations	Amount	Relative Growth from 2007, %
1	prognostic	271	4.00
2	prognosis	184	3.70
3	prognostic value	26	25.00
4	diagnosis prognosis	24	100.00
5	prognostication	22	100.00
6	prognostic factors	21	16.67
7	diagnostic prognostic	18	100.00
8	poor prognosis	14	50.00
9	prognostic model	12	100.00
10	independent prognostic	10	100.00
11	prognosis treatment	9	100.00
12	prognostic models	9	100.00
13	prognostic factor	9	100.00
14	prognostic index	7	100.00
15	prognostic stratification	7	100.00
16	prognostic score	6	100.00
17	prognoses	6	100.00
18	cancer prognosis	5	100.00

**Table 6 ijerph-17-07444-t006:** Terms and collocations with the root word “prevent” (yellow color in cyberspace).

No.	Terms and Collocations	Amount	Relative Growth from 2007, %
1	prevention	146	3.57
2	preventive	88	9.09
3	prevent	53	10.00
4	preventing	13	33.33
5	disease prevention	10	50.00
6	primary prevention	8	100.00
7	preventative	8	100.00
8	prevention treatment	7	100.00
9	preventive interventions	6	100.00
10	prevention strategies	6	100.00
11	melanoma-prevention	6	100.00
12	preventive measures	5	100.00
13	prevention management	5	100.00
14	stratified prevention	5	100.00

**Table 7 ijerph-17-07444-t007:** Combined use of significant terms with derivatives (predictor(s), predictive, predict(ing, ed), prediction) before and after 2007.

Word	Predictor(s)		Predictive		Predict(ing, ed)		Prediction	
	before	after	before	after	before	after	before	after
response	36	18	-	-	32	60	22	25
Significant, important	510	106	0	6	432	36	-	-
identify(ing, ied)	32	26	0	5	-	-	-	-
risk, error	-	-	-	-	0	24	189	77
model(s)	-	-	0	65	7	30	101	109
accur(acy, ate)	-	-	0	21	0	21	18	24
tool(s)	-	-	-	-	0	5	0	14
factor(s)	-	-	70	27	7	14	-	-
successful, reliable, efficacy	0	5	0	5	7	7	-	-
personalized, individual(ized )	-	-	0	13	28	14	0	28
disease, symptoms, diagnostic(s)	-	-	0	10	5	15	0	12
improve, develop(ment)	-	-	0	12	-	-	0	8
prognos(is, tic)	-	-	17	24	5	5	6	6
markers, biomarker(s)	-	-	131	62	0	16	-	-
unique(ly)	-	-	0	6	0	8	-	-
relationship	-	-	0	5	-	-	-	-
ability, capable, capability	-	-	0	5	0	15	-	-

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
