# Peer review of "Tracking Knowledge Evolution Based on the Terminology Dynamics in 4P-Medicine"

_ijerph, 2020, doi:10.3390/ijerph17207444_

Round 1

Reviewer 1 Report

This work could be considered as a good survey paper of the literature, I'm not sure if the authors has managed to consider the novelty of the work. The authors  are encouraged to write a section about the novelty of the work and a clear section shown the aims of the study, with a good plan of delivering the objectives. 

Author Response

Response to Reviewer 1 Comments

Point 1: Does the introduction provide sufficient background and include all relevant references?

Response 1: Lines 80-109. There were expanded introduction section.

Point 2: Is the research design appropriate?

Response 2: There were added lines 170-186.

Point 3: Are the methods adequately described?

Response 3: There were added lines 170-186.

Point 4: Are the conclusions supported by the results?

Response 4: We disagree, but the results and conclusion sections has been redesigned.

Point 5: This work could be considered as a good survey paper of the literature, I'm not sure if the authors has managed to consider the novelty of the work. The authors  are encouraged to write a section about the novelty of the work and a clear section shown the aims of the study, with a good plan of delivering the objectives.?

Response 5: We added information about the novelty (lines 164-168). We added the aim of the study (Line 25-26). Along with the description of the methods, we presented a good plan of delivering the objectives (lines 170-186).

Reviewer 2 Report

This paper built a collection of terms especially extracted from the PubMed database, revealed the temporal dynamics of the formation of derivatives and significant collocations of medical terms, and proposed special linguistic constructs. I believe this is a relevant study that is clearly connected to IJERPH and will be of interest to many readers.

However, there are a number of things that I am unclear about, that I think need to be addressed.

  1. I think the paper should define at the start what “4P medicine” is exactly. The term is not that common and it can help the reader to have a better sense of that the paper is by clearly defining the term from the start. Meanwhile, please unify the expression, authors sometimes write 4P medicine (e.g., line 3), sometimes “4P-medicine” (e.g., line 18).
  2. The paper needs to explain much clearer why you choose PubMed as it seems to be not enough to just say “we used a PubMed resource containing many scientific publications” (line 34), I suggest the authors introduce the database in the introduction.
  3. Please further discuss the theoretical and practical implications of the research, and explain the limitations as well as future directions of the paper
  4. I think the authors need to reframe the structure of 4 Discussion and 5 Conclusion section. Specifically, it would be better to put lines 261-287 and 294-304 in the Appendix rather than in 4 Discussion section; the research findings (for example, Figure 4-10, Table 7) in 5 Conclusion section should be categorized into 3 Results section,

In sum, I think the paper requires some further revisions to aid the reader in fully understanding what was accomplished.

Author Response

Response to Reviewer 2 Comments

Point 1: Does the introduction provide sufficient background and include all relevant references?

Response 1: Lines 80-109. There were expanded introduction section.

Point 2: Are the results clearly presented?

Response 2: We disagree, but the results section has been redesigned.

Point 3: Are the conclusions supported by the results?

Response 3: We disagree, but the results and conclusion sections has been redesigned.

Point 4: I think the paper should define at the start what “4P medicine” is exactly. The term is not that common and it can help the reader to have a better sense of that the paper is by clearly defining the term from the start. Meanwhile, please unify the expression, authors sometimes write 4P medicine (e.g., line 3), sometimes “4P-medicine” (e.g., line 18).

Response 4: We have redefined the term “4P-medicine” (lines 44-47).

Point 5: The paper needs to explain much clearer why you choose PubMed as it seems to be not enough to just say “we used a PubMed resource containing many scientific publications” (line 34), I suggest the authors introduce the database in the introduction.

Response 5: We described PubMed library and reasons for choosing it (lines 36-43).

Point 6: Please further discuss the theoretical and practical implications of the research, and explain the limitations as well as future directions of the paper.

Response 6: There is no limitations in this research. Future directions of research are presented in the conclusion (lines 473-479). Theoretical (lines 170-186) and practical (lines 470-472) implications of the research are presented.

Point 7: I think the authors need to reframe the structure of 4 Discussion and 5 Conclusion section. Specifically, it would be better to put lines 261-287 and 294-304 in the Appendix rather than in 4 Discussion section; the research findings (for example, Figure 4-10, Table 7) in 5 Conclusion section should be categorized into 3 Results section.

Response 7: Figure 4-10, Table 7 categorized into 3 Results section. Lines 261-287 transferred to the Appendix 1. Lines 294-304 transferred to the Appendix 2.

Point 10: In sum, I think the paper requires some further revisions to aid the reader in fully understanding what was accomplished.

Response 10: As a result of the revision, the article became more understandable for readers.

Reviewer 3 Report

This paper focus on 4p medicine and proposed a special linguistic constructs which can combining cross-lingual terms into a common semantic field. This paper used modern visualization technologies, and this method can help solve the problem of structuring multilingual heterogeneous information. The experimental content of this paper is rich, and the validity of the proposed method is proved by experiments on real data sets.The idea is innovative, the viewpoint is more interesting, has certain practical application value.

In general, the paper is clear in logic. It is suggested that this paper be accepted.

But there is some problems with this paper:

1)The experimental results in the figures can be appropriately described.

2)In the method description, the description of the technical method used should be more specific.

3)Summary part should adjust the order, you could put the idea of the paper at the end of a general summary.

4)There are some subtle grammatical problems.

Author Response

Response to Reviewer 3 Comments

Point 1: Is the research design appropriate?

Response 1: There were added lines 170-186.

Point 2: Are the methods adequately described?

Response 2: There were added lines 170-186.

Point 3: The experimental results in the figures can be appropriately described.

Response 3: Picture descriptions have been added (lines 52-54, 279-282, 291-294).

Point 4: In the method description, the description of the technical method used should be more specific.

Response 4:There were added lines 170-186.

Point 5: Summary part should adjust the order, you could put the idea of the paper at the end of a general summary.

Response 5: Line 25-26. We added the aim of the study.

Point 6: There are some subtle grammatical problems.

Response 6: As for grammatical problems, our article has been revised by a professional editor (Laurel Robinson Editorial Services, Machias, Maine, laurel@laurelcopyeditor.com).